# Iron Saturation Drives Lactoferrin Effects on Oxidative Stress and Neurotoxicity Induced by HIV-1 Tat

**DOI:** 10.3390/ijms24097947

**Published:** 2023-04-27

**Authors:** Giusi Ianiro, Veronica D’Ezio, Ludovica Carpinelli, Cecilia Casella, Maria Carmela Bonaccorsi di Patti, Luigi Rosa, Piera Valenti, Marco Colasanti, Giovanni Musci, Antimo Cutone, Tiziana Persichini

**Affiliations:** 1Department of Biosciences and Territory, University of Molise, 86090 Pesche, Italy; g.ianiro@studenti.unimol.it (G.I.); giovanni.musci@unimol.it (G.M.); 2Department of Science, University “ROMA TRE”, 00146 Rome, Italy; veronica.dezio@uniroma3.it (V.D.); ludovica.carpinelli@uniroma3.it (L.C.); cec.casella@stud.uniroma3.it (C.C.); marco.colasanti@uniroma3.it (M.C.); 3Department of Biochemical Sciences, Sapienza University of Roma, 00185 Rome, Italy; mariacarmela.bonaccorsi@uniroma1.it; 4Department of Public Health and Infectious Diseases, Sapienza University of Roma, 00185 Rome, Italy; luigi.rosa@uniroma1.it (L.R.); piera.valenti@uniroma1.it (P.V.)

**Keywords:** HIV-1 Tat, Lactoferrin, iron saturation, oxidative stress, neurotoxicity, astrocytes, Nrf-2, System X_c_^−^

## Abstract

The Trans-Activator of Transcription (Tat) of Human Immunodeficiency Virus (HIV-1) is involved in virus replication and infection and can promote oxidative stress in human astroglial cells. In response, host cells activate transcription of antioxidant genes, including a subunit of System X_c_^−^ cystine/glutamate antiporter which, in turn, can trigger glutamate-mediated excitotoxicity. Here, we present data on the efficacy of bovine Lactoferrin (bLf), both in its native (Nat-bLf) and iron-saturated (Holo-bLf) forms, in counteracting oxidative stress in U373 human astroglial cells constitutively expressing the viral protein (U373-Tat). Our results show that, dependent on iron saturation, both Nat-bLf and Holo-bLf can boost host antioxidant response by up-regulating System X_c_^−^ and the cell iron exporter Ferroportin via the Nuclear factor erythroid 2-related factor (Nrf2) pathway, thus reducing Reactive Oxygen Species (ROS)-mediated lipid peroxidation and DNA damage in astrocytes. In U373-Tat cells, both forms of bLf restore the physiological internalization of Transferrin (Tf) Receptor 1, the molecular gate for Tf-bound iron uptake. The involvement of astrocytic antioxidant response in Tat-mediated neurotoxicity was evaluated in co-cultures of U373-Tat with human neuronal SH-SY5Y cells. The results show that the Holo-bLf exacerbates Tat-induced excitotoxicity on SH-SY5Y, which is directly dependent on System-X_c_^−^ upregulation, thus highlighting the mechanistic role of iron in the biological activities of the glycoprotein.

## 1. Introduction

Human immunodeficiency virus (HIV)-1 infection is widely considered a public health problem. In 2021, nearly 1.5 million new infections and 650,000 deaths due to AIDS-related illnesses were recorded, and around 38.4 million individuals are estimated to live with HIV-1 infection (UNAIDS.org) [1].

Despite enormous advances in combined antiretroviral therapy (cART), the standard therapy for HIV-1 infection that significantly reduces viral load and prolongs life expectancy, HIV^+^ patients can experience HIV-associated neurocognitive disorders (HANDs), including reductions in total brain volume, thinning of cerebral cortex, and disturbances in functional connectivity [2].

Following infection, HIV enters the central nervous system (CNS) through circulating infected monocytes that cross the blood–brain barrier (BBB) and mediate the trans-infection of microglia and astroglia. These cells act as a reservoir for ongoing HIV-1 replication, thereby preventing the chance for a sterilizing cure or eradication. Therefore, although neurons are refractory to direct HIV infection, glial cells are involved in the establishing of HIV-1-related neurotoxicity.

As cART does not suppress the expression of HIV-1 non-structural proteins, the neurotoxic effects have been reported to be mainly linked to the chronic basal expression of different viral proteins, such as gp120, Nef (Negative factor), and Tat (Transactivator of transcription) [3]. Tat is known to be toxic to neurons, and it can be secreted from HIV-infected cells, including latently infected astrocytes [4]. This chronic low-level production of Tat has been proposed to contribute to neuronal damage over prolonged periods of time [5].

Some of the authors of this paper have recently shown that Tat induces intracellular oxidative stress in a human astroglial cell line, which in turn triggers a compensatory antioxidant response mediated by increased translocation in the nucleus of Nuclear factor erythroid 2-related factor (Nrf2) and the consequent transcriptional upregulation of antioxidant response element (ARE) genes, including Glutathione Peroxidase 4 (GPX4), Glutamate-cysteine ligase (GCL), and the xCT subunit of System X_c_^−^ cystine/glutamate antiporter (SLC7A11) [6]. The latter mediates the exchange of extracellular L-cystine and intracellular L-glutamate across the plasma membrane. While the import of cystine is necessary for intracellular glutathione production, the release of the neurotransmitter glutamate into the extracellular space can promote neurotoxicity [6]. Excitotoxicity occurs when neurons are exposed to high levels of glutamate that causes a persistent activation of different receptors, including the N-methyl-d-aspartate acid (NMDA) receptor, which in turn results in a lethal influx of extracellular calcium, thus leading to cell death via the generation of free radicals [7,8].

In this context, iron plays a fundamental role as a major actor in the promotion and exacerbation of oxidative stress. The changeable nature of iron from ferrous (Fe^2+^) to ferric (Fe^3+^) ion, and vice versa, makes it critical for biological processes whilst rendering it very reactive against oxygen, thus promoting reactive oxygen species (ROS) production. Therefore, iron homeostasis needs to be strictly regulated at the cellular and systemic level, and its alteration has a role in a wide range of diseases, including those affecting the CNS [9,10]. Briefly, iron entry in human cells is mainly guaranteed by the binding of Transferrin (Tf), the main protein involved in systemic iron transfer, to its receptor (TfR1) through an endosome-mediated intake. In the acidic environment of the endosome, iron is released from Tf and translocated by Divalent Metal Transporter 1 (DMT1) into the cytoplasm, where, if in excess, it is sequestered by Ferritin (Ftn), the main iron storage protein. At occurrence, the metal is exported by Ferroportin (Fpn), the only known cell-to-blood iron transporter in vertebrates, mainly expressed by enterocytes, hepatocytes, macrophages [11], and astroglial cells [12]. Fpn works in conjunction with an iron oxidase, the nature of which depends on the cell type: Hephaestin is found mainly in epithelial cells and Ceruloplasmin (Cp) in most other cell types, including astrocytes. In the brain, the elevated iron demand for mitochondrial electron transport, myelin formation, and for metabolism of catecholamine neurotransmitters needs a fine regulation of iron influx [13]. Once iron has entered the BBB and cerebrospinal fluid barrier, it is taken up by astrocytes and made available to neuronal cells. In astrocytes, neurons, and microglia, the main iron uptake system involves Tf/TfR1, while iron release is carried out by the Fpn/Cp couple [14]. Within this frame, astrocytes act as the storage and regulation system of cerebral iron homeostasis and exert a protective role towards neurons by reducing the free iron pool in the synaptic cleft [14]. Although brain cells tightly control iron entry, numerous studies have evidenced that iron-induced oxidative stress can nevertheless occur and contributes to the onset and maintenance of neurodegenerative diseases, such as Alzheimer’s, Parkinson’s, and Friedreich’s ataxia, as well as inflammatory and infectious pathological conditions [15]. Indeed, iron homeostasis is perturbed during HIV viral infection, when pro-inflammatory cytokines worsen the pathological state, leading to both higher intracellular iron content, which facilitates viral spreading [16], and ROS production, thus globally promoting cell damage and organ failure, including neuropathologies [17].

In this scenario, bovine Lactoferrin (bLf), a milk-derived iron-binding glycoprotein, could play a pivotal protective role. It exerts pleiotropic functions, from anti-microbial, to anti-viral, anti-inflammatory, and antioxidant ones [18,19,20,21]. In almost all studies, bLf has been used as bioequivalent of human Lf by virtue of high sequence homology and striking functional sharing. In addition, it is classified as a “generally recognized as safe” (GRAS) substance by the USA Food and Drug Administration, which has made it a readily available nutraceutical. Depending on its iron content, Lf is present in native (Nat-Lf) and Holo (Holo-Lf) forms, with an iron saturation rate of 10–20% and more than 95%, respectively. In this regard, recent efforts are being made to investigate the differential biological activities between the Nat- and Holo-Lfs. Notably, Lf is able to maintain the physiological balance of ROS levels by both binding free ferric ions and regulating the main enzymes involved in cellular antioxidant response, such as SOD and GPX [20]. Moreover, Lf is emerging as a modulator of iron homeostasis [22]. In recent years, our group has demonstrated the efficacy of bLf in reverting iron dysregulation in different models of inflammation/infection, both in vitro [23,24] and in vivo [25], as well as in clinical trials [26,27]. The effect of Lf on iron homeostasis can be related to its ability to chelate free iron, modulate iron proteins, and downregulate pro-inflammatory cytokines, such as IL-1β and IL-6, thus boosting antioxidant and anti-inflammatory host response, also after viral infection.

Here, for the first time, we present data on the effects of both Nat-Lf and Holo-bLf on HIV-1 Tat-induced oxidative stress and neurotoxicity in in vitro models.

## 2. Results

### 2.1. Native and Holo-Bovine Lactoferrin Are Internalized by U373 Cells

Depending on the biological system, Lf has been demonstrated to exert its multiple functions by interacting with different cell surface receptors [28], leading either to intracellular signaling cascades or to receptor-mediated endocytosis and translocation to the cell nucleus. Hence, we first analyzed Lf internalization and subcellular localization in U373 cells, either expressing the Tat protein or not. No expression of endogenous Lf in these cells was detected. After 24 h of treatment, both Nat- and Holo-bLf were efficiently internalized by U373 cells, independently from Tat expression (Figure 1). bLf targeted both nuclear and cytoplasmic compartments, with higher levels of Holo-bLf detected in both U373 and U373-Tat cells.

### 2.2. Bovine Lactoferrin Modulates Cell Antioxidant Response

We then evaluated the effect of bLf towards the compensative cell response against oxidative stress induced by Tat. For this purpose, the nuclear translocation of the transcription factor Nrf2 (Figure 2) and the expression of System X_c_^−^ (Figure 3 and Figure 4) were analyzed.

As shown in Figure 2, the basal level of activated Nrf2 is 50% higher in U373-Tat compared to U373-mock. Irrespective of Tat expression, Lactoferrin enhanced the amount of activated Nrf2, with Holo-bLf being invariably more efficient than Nat-bLf. The effect of the iron glycoprotein was additive to that exerted by Tat. In the presence of both Tat and Holo-bLf, up to a 3.3-fold increase in Nrf2 transcription factor nuclear levels was attained (Figure 2).

Real-Time qPCR analysis on total RNA extracts was carried out to evaluate System X_c_^−^ gene expression. Consistent with enhanced Nrf2 nuclear translocation, a 2-fold increase in System X_c_^−^ was recorded in U373-Tat cells as compared to U373-mock, as previously reported [6] (inset of Figure 3C). The time-dependency of the effect of Nat- or Holo-bLf on U373-Tat and U373-mock cells is depicted in Figure 3A–D. As observed for Nrf2 nuclear translocation, bLf was always able to enhance the transcription of System X_c_^−^, its effect peaking at 4–8 h after treatment, with minor differences depending on the iron saturation rate and on the presence of Tat.

To corroborate the expression analysis on transcript levels, protein expression was evaluated as well. Western blots confirmed the increment of System X_c_^−^ expression in U373-Tat cells with respect to the control cells [6] and the boosting effect exerted by bLf which, globally, acts as a positive regulator for System X_c_^−^ (Figure 4). Consistent with the transcriptional data of Figure 3, a remarkable up-regulation of System X_c_^−^ production was obtained upon Holo-bLf treatments in both U373 and U373-Tat cells.

### 2.3. Bovine Lactoferrin Potentiates Cell Defense against Intracellular Iron Overload

Next, given the tight correlation between oxidative stress and iron/inflammatory disorders, we evaluated the expression of systems involved in iron homeostasis.

We first measured levels of Ftn, the iron storage protein, and of nuclear receptor coactivator 4 (NCOA4), a cargo receptor for the selective turnover of Ftn. As shown in Figure 5A,B, U373 cells expressing Tat displayed a significant reduction in Ftn levels and the concomitant up-regulation of NCOA4, suggesting a possible activation of ferritinophagy. Neither Nat- nor Holo-bLf affected the process.

We then moved to the systems involved in iron intake (TfR1) and export (Fpn). Tat-expressing cells showed a down- and up-regulation, respectively, compared to control cells. Holo-bLf significantly potentiated these effects for both Fpn and TfR1, while the Nat-form was only effective on Fpn levels (Figure 5C,D). In addition, levels of Cp, the ferroxidase partner of Fpn, decreased in the presence of the viral protein (Figure 5E). Of note, Holo-bLf reduced Cp expression in U373-mock cells, whereas both Nat- and Holo-bLf did not revert the effect of Tat.

Concerning IL-6, its production did not change in Tat vs. mock cells, whereas treatment with Holo-bLf promoted a significant IL-6 release in both U373-Tat and U373-mock cells (Figure 5F). Levels of IL-1β were below the detection threshold in all experimental conditions tested.

Confocal microscopy revealed a perinuclear localization of both Ftn and TfR1 in control cells, which is not influenced by bLf treatments, neither in Nat- nor Holo forms (Figure 6). Interestingly, a more spread out and membrane-linked signal of TfR1 was found in Tat-expressing U373, suggesting an impairment of receptor internalization, whereas bLf treatments seem to re-localize TfR1 signals to the perinuclear compartment. Concerning Ftn, the signals in bLf-treated U373-Tat cells were reduced with respect to control cells, in accordance with Western blot data (Figure 5A).

Overall, our data suggest that bLf, mainly in the Holo-form, counteracts intracellular free metal accumulation, possibly associated with Tat-induced ferritinophagy, by enhancing Fpn-mediated iron release and reducing Tf-bound iron uptake, thus limiting the excess of intracellular pro-oxidant iron.

### 2.4. Bovine Lactoferrin Counteracts Tat-Mediated Lipid Peroxidation and DNA Damage in Astroglial Cells

To verify whether bLf protects U373 cells from intracellular ROS accumulation, lipid peroxidation measurements were carried out through the BODIPY test.

As shown in Figure 7A, Tat expression promoted a significant increase in lipid peroxidation compared with control cells. Both Nat- and Holo-bLf efficiently counteracted the increase and restored physiological lipid peroxidation levels. Moreover, bLf treatments were also able to significantly decrease peroxidation in U373 cells, thus suggesting a protective role against ROS accumulation, independently from Tat expression.

Consistent with the BODIPY analysis, a significant increase in histone variant γ-H2AX, a selective marker of DNA damage, was recorded in Tat-expressing cells (Figure 7B). Again, bLf reverted Tat-induced up-regulation of γ-H2AX, thus suggesting a potential protective effect on DNA damage.

### 2.5. Holo-bLf Exacerbates Tat-Induced Neutoxicity via System X_c_^−^

Given the tight interconnection between astroglial metabolism and neuronal functions, co-cultures of U373, both in the parental and Tat-expressing phenotype, and SH-SY5Y neuronal cells were set up.

Co-cultures were treated with Nat- and Holo-bLf at a concentration of 100 μg/mL for 24 h and viability tests through both MTT assay and cell counts were carried out.

As shown in Figure 8A, there is a significant reduction in the viability of neurons co-cultured with U373-Tat compared to control, in agreement with our previous report [6]. Lf, both in its Nat- and even more in its Holo-form, induced a further significant reduction in neuronal viability, both in the presence and absence of Tat expression.

Essentially the same results were obtained through cell counts (Figure 8C). On the other hand, Lf treatment did not affect astroglial cells viability in co-cultures either in the presence or absence of Tat expression (Figure 8B,D).

In order to elucidate the molecular mechanism underlying the reduction in SH-SY5Y viability in co-cultures, the experiments were repeated in the presence of N-acetylcysteine (NAC), a well-established scavenger of ROS. The MTT assay was performed following a 30 min pre-treatment with NAC before a 24 h incubation with Nat-bLf or Holo-bLf. The results unequivocally show that NAC can fully restore the viability of neurons in co-cultures with either U373-mock or U373-Tat (Figure 9A). As expected, the viability of astroglial cells in co-culture was not affected by pre-treatment with NAC either in the presence or absence of Tat (Figure 9B).

To test the involvement of System X_c_^−^ in neuronal toxicity, parallel sets of experiments were carried out in the presence of sulfasalazine (SSZ), a specific System X_c_^−^ inhibitor. Again, MTT assays were performed on cells pre-treated with the inhibitor for 30 min and then incubated for 24 h with Nat-bLf or Holo-bLf. As shown in Figure 9C, the viability of neurons co-cultured with U373-mock or U373-Tat was invariably restored by pre-treatment with SSZ under all conditions tested (Figure 9C). Quite interestingly, while the inhibitor did not affect viability of astroglial U373, both in the presence and absence of bLf, it significantly reduced U373-Tat viability by about 30% compared with controls, and the presence of bLf in the native form restored the viability of U373-Tat to the control level (Figure 9D).

To further decipher the molecular mechanism, we measured the amount of glutamate released by astrocytes into the culture medium. The glutamate release was normalized on the number of viable astrocytes and the ratio was arbitrarily set to 1 for co-cultures with untreated U373-mock cells. Consistent with the above data on viability and System X_c_^−^ expression, Holo-bLf induced a marked increase in glutamate release in co-cultures with both U373-mock cells (Figure 10A) and U373-Tat cells (Figure 10B), whereas Nat-bLf showed a slight increase only in co-culture with U373-Tat cells (Figure 10B).

Treatments and co-treatments with SSZ drastically decreased glutamate release in all tested conditions, thus further demonstrating the role of System X_c_^−^ in promoting neuronal excitotoxicity (Figure 10A,B).

Moreover, viability experiments on co-cultures of SH-SY5Y and U373-Tat cells were carried out in the presence of MK801, a specific antagonist of NMDA receptors. As depicted in Figure 11A,B), while astroglial cells were unaffected, the significant reduction in neuronal viability induced by Holo-bLf treatment is fully reverted by co-treatment with the NMDA antagonist, thus further corroborating the role of System X_c_^−^-mediated glutamate release in excitotoxicity.

Finally, Holo-bLf induced a significant increase in lipid peroxidation in SH-SY5Y cells co-cultured with U373-Tat, whereas Nat-bLf did not affect neuronal lipid peroxidation with respect to the control (Figure 12A), thus supporting a role for iron in astrocyte-mediated ROS generation and excitotoxicity. Interestingly, consistent with the role of the astrocytic antioxidant response, neither the Holo-Lf nor the Nat-Lf were able to induce lipid peroxidation in SH-SY5Y cell monocultures (Figure 12B).

## 3. Discussion

Viruses rely on host factors for initial access, cell invasion, and replication processes. Among them, RNA viruses show the highest mutation rates among all living organisms [29], thereby limiting the development of effective vaccines and drugs. Several studies have suggested that RNA viruses, upon entry into the host, exploit the environment induced by oxidative stress for genome capping and replication, thus contributing to disease severity [30]. Oxidative stress invariably plays a dominant pathogenic role in HIV-1 infection, and parallel activation of host antioxidant pathways orchestrated by the transcription factor Nrf2 contributes to the regulatory control of antiviral and apoptotic responses by maintaining redox homeostasis [31]. In agreement, a growing body of evidence suggests that viruses exploit intracellular iron for their own replication [16,32]. HIV-1 infection sets the body into a systemic inflammatory state leading to intracellular iron load and anemia, which have been associated with increased viral replication and worse prognosis in HIV^+^ patients, respectively [33,34]. In this respect, iron chelators could represent a promising adjuvant therapy in HIV^+^ patients as they can dramatically reduce oxidative stress and suppress viral survival [35]. Moreover, increased iron export through Fpn has been directly associated with restriction of HIV-1 infection [36].

Numerous works on HIV^+^ models have highlighted the role of the viral proteins as neurotoxic factors that, interacting directly or indirectly with neurons, can induce oxidative stress and neuroinflammation [37,38]. Indeed, we reported that HIV-1 Tat can induce neurotoxicity by eliciting spermine oxidase-dependent ROS generation through the stimulation of the NMDA receptor in SH-SY5Y cells, which in turn leads to GSH depletion and oxidative stress [39].

However, the interplay between oxidative stress and iron dysregulation in HIV-associated neurocognitive disorders is still poorly understood.

Here, we present data on the antioxidant cell response in human astroglial cells constitutively expressing and secreting the viral protein Tat, thus simulating a condition occurring in seropositive patients, such as astrocyte infection. The activation of the Nrf2 pathway is observed, along and consistent with up-regulation of System X_c_^−^, both at transcriptional and translational levels. As far as iron proteins are concerned, our data are consistent with the activation of ferritinophagy. Within this frame, up-regulation of Fpn and down-regulation of TfR1 and Cp are recorded in Tat-expressing cells compared to the control. Moreover, in line with studies on HIV-1 Nef protein [40], we have shown that Tat protein impairs the internalization of TfR1, thus possibly impairing HIV-1 entry into the host cell. It should be noted that TfR1 has been described as a secondary gate for a plethora of viruses [41], including the most recent SARS-CoV-2 [42].

Overall, our data are supportive of a cell response by astrocytes towards Tat-induced oxidative stress by counteracting intracellular ROS accumulation and free iron content. However, this response seems to fail in preventing ROS-mediated macromolecular damage, as evidenced by the significantly higher value of lipid peroxidation and DNA damage registered for U373-Tat cells as compared to the control.

On the other hand, bLf, both in Nat- and Holo-forms, significantly reduced lipid peroxidation and, for the sole Nat-bLf, DNA damage. bLf, depending on iron content, has been shown to potentiate the antioxidant response of astroglial cells by both up-regulating System X_c_^−^ and decreasing intracellular iron overload via Fpn-mediated export. In Tat-expressing cells, bLfs could restore physiological recycling of TfR1 while, only for the Holo form, potentiate Tat-induced receptor downregulation, as evidenced by the Western blot. Notably, while Holo-bLf was more efficient in promoting the expression of System X_c_^−^ and Fpn, Nat-bLf down-regulated histone variant γ-H2AX and did not induce IL-6 expression. These dissimilar outcomes may be directly related to the glycoprotein iron content, which is strictly connected to differences in structural and functional features. As demonstrated by the subcellular localization experiments, Holo-bLf accumulates more efficiently and is more retained into the cell nucleus than its native counterpart, thus suggesting a more prolonged and capable regulation of gene expression. Similar effects have been described in our recent study on a model of human glioblastoma [43]. This ability is easily explained by the higher resistance to proteolysis reported for Holo-Lf when compared to the totally iron-depleted (Apo) and Nat-forms [44]. However, unlike the iron-saturated form, whose effectiveness seems mainly related to the up-regulation of System X_c_^−^ and Fpn, Nat-bLf can act directly as an ROS scavenger by chelating reactive free iron, thus counteracting the action of pro-oxidant species.

In the brain, HIV-1-infected astrocytes and microglia may play a detrimental role, being able to trigger excessive release of glutamate and several neurotoxic factor such as proinflammatory cytokines and nitric oxide which, in turn, promotes ROS-mediated neurotoxicity [45]. Glutamate release via System X_c_^−^ from both microglia and astrocytes has been reported to enhance the excitotoxicity of cortical neurons [46,47]. Additionally, some of the authors of the current paper have previously found that System X_c_^−^ contributes to increased glutamate excitotoxicity in the neocortex of a mouse model (Dach-SMOX) showing constant and chronic oxidative stress [48]. One of the earliest pieces of evidence on the involvement of System X_c_^−^ as a possible source of excitotoxic glutamate comes from a study on microglia expressing high levels of the transporter due to the sustained need for oxidative protection [49]. More recently, it has been found that Tat elicits microglial glutamate release via System X_c_^−^, thus suggesting that Tat-induced extracellular glutamate increase might contribute in part to neurologic disfunctions associated with HIV infection [50]. In astrocytes, System X_c_^−^ can be up-regulated by interleukin-1β, thus leading to enhanced hypoxic neuronal injury [46]. Furthermore, we previously reported that the expression of IL-1β as well as of nitric oxide synthase was elicited in astrocytes by HIV gp120, thus fueling a vicious circle [38]. It is interesting to note how inflammatory pathways and oxidative stress may converge to System X_c_^−^ activation, suggesting a possible mechanism underlying HIV-induced excitotoxicity. In this respect, it should be recalled that HIV^+^ patients have levels of glutamate fivefold greater than healthy controls in their cerebrospinal fluid (CSF) [51]. Recent studies have demonstrated that HIV^+^ patients administered with cART and suffering from HIV-associated neurocognitive disorders show significant increases in CSF levels of glutamate compared with patients without neurocognitive impairment [52], whereas lower levels of the neurotransmitter are found in the parietal grey matter, basal ganglia, and cortex [53]. These results are consistent with a concomitant malfunction of glutamate recycling via the glutamate–glutamine shuttle, leading overall to increased glutamate levels in the extracellular space.

Given the involvement of astrocytes in CNS pathology, it is not surprising that the ability to exacerbate neurodegeneration through the conversion of oxidative stress to excitotoxicity via System X_c_^−^ has been linked to a variety of disorders, including Alzheimer’s disease, Parkinson’s disease, AIDS, multiple sclerosis, and ALS [54]. In this respect, we have recently reported that Amyloid-β peptide can induce neurotoxicity through the up-regulation of astrocytic System X_c_^−^ [55].

As a whole, System X_c_^−^ has become a new potential pharmacological target for brain neurodegenerative diseases associated with an excess of extracellular glutamate. Unfortunately, no selective System X_c_^−^ inhibitor has been shown to be efficient and safe in clinical application. Here, we demonstrated how Holo-bLf, by acting as a positive regulator of the Nrf2 pathway, is able to induce System X_c_^−^ over time, thereby potentiating its excitotoxic effects against the SH-SY5Y neuronal cells. It is tempting to speculate a dual effect of bLf depending on its chemical formulation: on one hand, bLf leads to a deleterious exacerbation of neurodegeneration, and on the other hand it may offer therapeutic benefits (e.g., in preventing the onset of amyloidosis [56]).

These results highlight, once again, the importance of quality standardization for the numerous commercial preparations of bovine Lactoferrin, which, to date, are totally lacking. Although bLf is classified as a dietary supplement, a critical point is that there are currently several bLf products on the market that show high variability and discrepancy in their effects. Until the FDA and EFSA decide to make bLf quality analysis mandatory, there will be products that do not conform to their claimed functions. In this regard, iron content has been only partially investigated as a key parameter that drives the glycoprotein results depending on the system it acts upon. In the present study, we show how Nat-bLf can protect astroglial cells from Tat-induced oxidative stress and sequelae, but without inducing excitotoxicity. On the other hand, Holo-bLf, although able to induce a greater antioxidant response, failed in protecting the host from DNA damage, up-regulated the pro-inflammatory cytokine IL-6, and significantly exacerbated Tat-induced neurotoxicity via System X_c_^−^, thus acting as a double-edged sword. Although Lf has been shown to act as a potent neuroprotective agent [56,57]**,** and no evidence of bLf-induced neurotoxicity has ever been reported in vivo either in animal models or in clinical trials, our study highlights the utmost importance of considering Lf iron saturation when applied to clinical therapies, as it can dramatically influence the success or failure of the treatment itself.

## 4. Materials and Methods

### 4.1. Reagents

DMEM (Dulbecco’s modified Eagle’s medium), FBS (Fetal Bovine Serum), Trypsin–EDTA 0.25% solution, gentamicin 50 mg/mL solution, G418 (Geneticin), sulfasalazine (SSZ; a specific inhibitor of System X_c_^−^), MK-801 hydrogen maleate (MK-801; an NMDA receptor antagonist), N-Acetyl-L-cysteine (NAC), and the kit for MTT assay were obtained from Sigma–Aldrich (Milan, Italy). The reagent for the Bradford assay was from Bio-Rad Italia (Milan, Italy). All chemicals were of reagent or analytical grade and were used without further purification. TRIzol Reagent was from Life technologies Italia-Invitrogen, (Monza, Italy). The Go Taq 2-Step RT-qPCR System kit was obtained from Promega Italia Srl, (Milan, Italy) and the SsoAdvanced universal SYBR green supermix was from Bio-Rad Italia (Milan, Italy).

### 4.2. Bovine Lactoferrin

Highly purified bLf (Saputo Dairy, Southbank, VC, Australia) was generously supplied by Vivatis Pharma Italia s.r.l. Protein purity was about 99%, as checked by SDS-PAGE and silver nitrate staining. The concentration of bLf solutions was assessed via UV spectroscopy with an extinction coefficient of 15.1 (280 nm, 1% solution). Iron saturation was about 11%, as determined via optical spectroscopy at 468 nm using an extinction coefficient of 0.54 for a 1% solution of 100% iron-saturated protein. LPS contamination, assessed via Limulus Amebocyte assay (Pyrochrome kit, PBI International, Milan, Italy), was 0.5 ± 0.06 ng/mg. Before each in vitro assay, bLf solutions were sterilized using a 0.2 μm Millex HV filter at low protein retention (Millipore Corp., Bedford, MA, USA).

Holo-bLf was prepared by incubating native bLf (20 mg/mL in 0.1 M sodium bicarbonate) with 5 mM ferric citrate for 2 h under stirring. The resultant Holo-bLf was then dialyzed against 0.1 M sodium bicarbonate for 48 h to remove unbound iron. The obtained Holo-bLf, >95% iron-saturated, was frozen and stored at −20 °C prior to experimental usage.

### 4.3. Cell Cultures, Transfection and Treatments

Human glioblastoma astrocytoma cells (U373-MG) and human neuroblastoma cells (SH-SY5Y) were purchased from American Type Culture Collection (ATCC, Manassas, VA, USA). Both cell lines were maintained in culture at 37 °C, in 5% CO_2_ atmosphere, using DMEM supplemented with 10% FBS, 2 mM L-glutamine, 40 μg/mL gentamicin. Confluent monolayers were sub-cultured by conventional trypsinization.

U373-MG cells were transfected with pcDNA3.1 (U373-mock) or pcDNA3.1-HIV-Tat (U373-Tat) expression vectors as previously reported [6,39]. For the maintenance of transfected cells in culture, G418 (200 μg/mL) was added to the culture medium.

Treatments with Nat-Lf (100 μg/mL) and Holo-bLf (100 μg/mL) were performed in DMEM without FBS for different time as indicated. Pre-treatments with Sulfasalazine (300 μM), NAC (2 mM) or MK-801 (10 μM), where indicated, were performed 30 min before the addition of bLf to the cell cultures. All experiments reported in this study were repeated at least three independent times.

### 4.4. Co-Cultures and Cell Viability Assay

SH-SY5Y cells were grown in co-cultures with U373-mock or U373-Tat using a transwell culture system as previously reported [55]. For each sample in co-cultures, 1 × 10^5^ neuronal cells were seeded in transwell insert and 2.5 × 10^5^ astroglial cells were plated in the lower compartment of a 6-well plate and allowed to grow for 24 h. U373 and SH-SY5Y cell viability was assessed with MTT assay at the end of the incubation period. MTT solution (0.5 mg/mL MTT dissolved in PBS) was added to the cell culture at the final concentration of 10%, and the cells were left in the incubator at 37 °C for 4 h. Formazan crystals were dissolved in MTT solvent (4 mM HCl, 0.1% NP40 in isopropanol), and samples were incubated at 37 °C for 30 min. The optical density (OD) of each sample was then measured at a wavelength of 570 nm using the Tecan Spark10M reader (Tecan, Mannedorf, Switzerland). In some experiments, where indicated, the Trypan blue exclusion assay was used to quantify viable cells.

### 4.5. Quantitative Real-Time Reverse Transcription-Polymerase Chain Reaction

Total RNA was purified by using Trizol Reagent and reverse transcribed into cDNA with GoTaq 2-step RT-qPCR system. cDNA was then amplified for System X_c_^−^ gene (xCT subunit, NM_014331.4) and Glyceraldehyde 3-phosphate dehydrogenase (GAPDH, NM_002046.7) mRNA was examined as the reference cellular transcript. The sequences of primers were as previously reported [55]. PCR products were quantified using the SYBR-Green method. Reactions were performed in an Agilent Aria Mx machine (Agilent technologies, Santa Clara, CA, USA) using the following program: 45 cycles of 95 °C for 15 s, 60 °C for 60 s, 72 °C for 20 s. GAPDH mRNA amplification products were present at equivalent levels in all cell lysates. Values were calculated relative to the internal housekeeping gene according to the second derivative test (delta–delta Ct (2^−ΔΔCT^) method).

### 4.6. Total and Nuclear Extracts

Preparation of total extracts was performed by adding 1% TEEN triton buffer (10 mM tris HCl pH 7.4, 1 mM EDTA, 1 mM EGTA, 150 mM NaCl, 1% Triton X-100, protease inhibitor cocktail dissolved 1:100) to the cell pellets. Samples were kept at 4 °C for 20 min, and shaken every minute. Subsequently, samples were centrifuged at 14,000 rpm, 4 °C for 20 min, and then the supernatant containing the cellular proteins was removed and stored in aliquots at −80 °C.

Preparation of nuclear extracts was performed by adding buffer A (10 mM Hepes pH 7.9, 10 mM KCl, 1.5 mM MgCl_2_, 0.5 mM DTT, 0.1% NP40, protease inhibitor cocktail dissolved 1:100) to the cell pellets to separate nuclei from cytosol. After incubation for 10 min on ice, samples were centrifuged at 12,000 rpm for 10 min at 4 °C. Thereafter, pellets containing nuclear fractions were resuspended in buffer C (20 mM Hepes pH 7.9, 420 mM NaCl, 1.5 mM MgCl_2_, 25% glycerol, 1 mM EDTA, 1 mM EGTA, 0.5 mM DTT, 0.05% NP40, protease inhibitor cocktail dissolved 1:50) and incubated on ice for 30 min. A final centrifugation at 14,000 rpm was carried out, and the supernatants were collected and stored at −80 °C.

The total protein content was determined according to the Bradford method [58].

### 4.7. Western Blotting

Proteins were separated by SDS-PAGE and electroblotted onto nitrocellulose (GE Healthcare, Life Sciences, Little Chalfont, Buckinghamshire, UK). Following transfer, each membrane was incubated in TBS-T (Tris Buffer Saline: 20 mM Tris-HCl pH 7.4; 137 mM NaCl; 0.1% Tween 20) containing 5% non-fat dried milk powder (Blotting-Grade Blocker, PanReac AppliChem, ITW reagents, Monza, Italy) for 1 h at room temperature. The membrane was incubated overnight at 4 °C with primary antibody dissolved in TBS-T containing 5% milk.

The following primary antibodies were employed: polyclonal anti-actin (A2066 Sigma-Aldrich; Milan, Italy) (1:1000), polyclonal anti-System X_c_^−^ (Ab175186 Abcam, Milan, Italy) (1:1000), polyclonal anti-Nrf2 (16396-1-AP Protein Tech; Manchester, United Kingdom) (1:1000), polyclonal anti-Ftn (sc25617, Santa Cruz, CA, USA) (1:1000), polyclonal anti-HCP (A0031, Dako, Santa Clara, CA, USA) (1:1000), polyclonal anti-lamin A (Ab26300 Abcam; Milan, Italy) (1:1000), all of them of rabbit origin; monoclonal anti-vinculin (sc-73614, Santa Cruz, CA, USA) (1:1000), monoclonal anti-ARA70 (NCOA4) (sc-373739, Santa Cruz, CA, USA) (1:1000), monoclonal anti-bLf (sc-53498, Santa Cruz CA, USA) (1:1000), monoclonal anti-Fpn 31A5 (1:1000), generously provided by T. Arvedson (Amgen), monoclonal anti-TfR1 (sc-32272, Santa Cruz, CA, USA) (1:1000), and monoclonal anti- *p*-Histone H2A.X (Ser 139) (sc-517348, Santa Cruz, CA, USA) (1:1000), all of them of mouse origin.

The membrane was then incubated with the appropriate HRP-conjugated secondary antibody (Biorad, Milan, Italy) (1:1000) in TBS-T containing 2.5% milk for 1 h at room temperature. The reagent used for detection was Clarity Western ECL substrate (170-5061, Biorad, Milan, Italy).

Differences between samples with respect to the presence of the proteins of interest were normalized using actin protein for total extracts and lamin A protein for nuclear extracts as reference.

### 4.8. Cytokine Analysis

Quantitation of IL-6 and IL-1β was performed on cell supernatants with ELISA, using Human ELISA Max Deluxe Sets (BioLegend, San Diego, CA, USA).

### 4.9. Immunocytochemistry and Confocal Analysis

Cells were grown on coverslips and fixed with 4% PFA in PBS, followed by permeabilization with 0.1% Triton X-100 in PBS. TfR1 and Ftn primary antibodies (1:100) were incubated overnight at 4 °C and visualized by means of Alexa Fluor (Invitrogen, Carlsbad, CA, USA). Coverslips were stained with the fluorophore-conjugated secondary antibodies (Alexa Fluor™ 546 and Alexa Fluor™ 488, Invitrogen, Carlsbad, CA, USA) and Hoechst for nuclei visualization, were mounted in antifade (SlowFade; Invitrogen, Carlsbad, CA, USA), and examined under a confocal microscope (TCS SP8; Leica, Wetzlar, Germany) equipped with a 40 × 1.40–0.60 NA HCX Plan Apo oil BL objective at RT.

### 4.10. Measurements of Glutamate Concentration in Cell Supernatants

To evaluate glutamate release in the supernatants of co-cultures, Glutamate Assay (Abcam, Milan, Italy) was performed as indicated by manufacturer’s instructions. Briefly, 20 μL of each sample supernatant were collected in a 96-well plate and assay buffer was added up to 50 μL final volume. Then, 100 μL of the reaction mix were added to each well, the plate was incubated for 30 min at 37 °C, protected from light, and optical density (OD) was measured at 450 nm in a microplate reader. Glutamate concentration of each sample was calculated using glutamate standard curve (0, 1.3, 6.5, 13, 26, 40, 53, 67 μM).

### 4.11. Measurements of Lipid Peroxidation

To detect lipid peroxidation in U373 and/or SH-SY5Y cells, the fluorescence probe C11-BODIPY 581/591 (D3861, Thermo Fisher Scientific, Waltham, MA, USA) was used. In reduced state, the excitation and emission maxima of C11-BODIPY 581/591 is 581/591 nm; after oxidation, the probe shifts the excitation and emission to 488/510 nm. Briefly, the cells of each sample were incubated with C11-BODIPY 581/591 diluted in DMEM without FBS for 45 min at 37 °C. Subsequently, cells were fixed in PFA 4% and observed under a confocal microscope. The green and red fluorescence signals were acquired simultaneously using double wavelength excitation and detection. The quantification of lipid oxidation was calculated by measuring the fluorescence emitted by the C11-BODIPY 581/591 probe, using Image J software (LOCI, University of Wisconsin, Madison, WI, USA) and the formula: G/G+R (G: Green; R: Red). After setting to 1 arbitrary unit the value of the control cells, values for the other samples were calculated relative to it.

### 4.12. Statistical Analysis

All data are expressed as mean ± standard error of the mean (SEM) of n observations. Statistical analysis of the data was performed using Graph Pad PRISM software (San Diego, CA, USA). Statistical significance was assessed with the one-way ANOVA test, followed by the Tukey post-test. Differences are considered statistically significant at *p* ≤ 0.05.

## Figures and Tables

**Figure 1 ijms-24-07947-f001:**
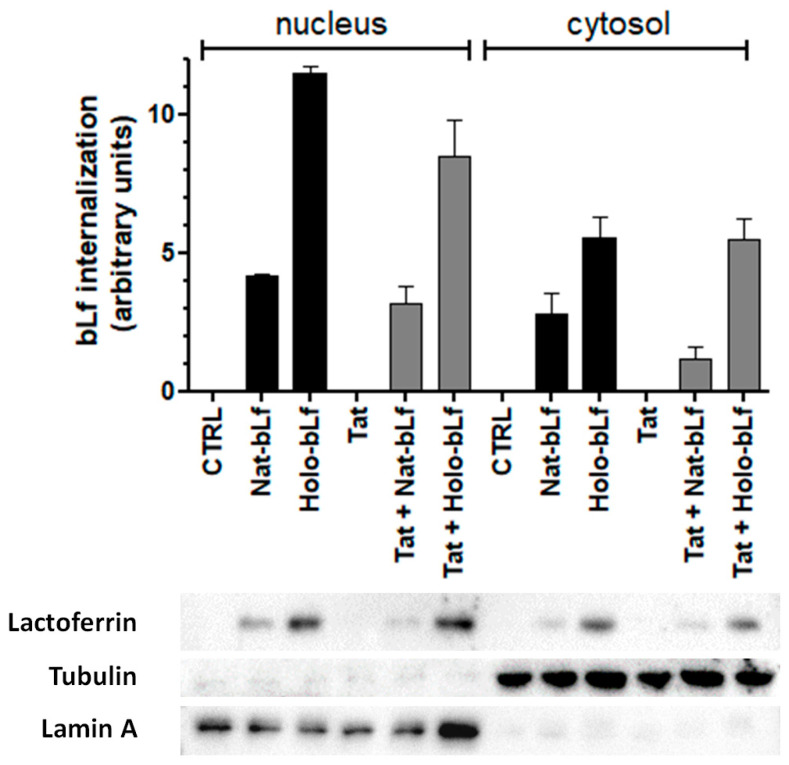
Analysis of bLf internalization and subcellular localization in U373 and U373-Tat cells. Western blot and densitometric analysis of bLf in nuclear and cytosolic fractions after 24 h of treatment with 100 μg/mL of Nat-bLf or Holo-bLf. Data are calculated relative to the internal housekeeping gene (Lamin A for nuclear fraction and Tubulin for cytosolic fraction) and are expressed as the means ± SEM.

**Figure 2 ijms-24-07947-f002:**
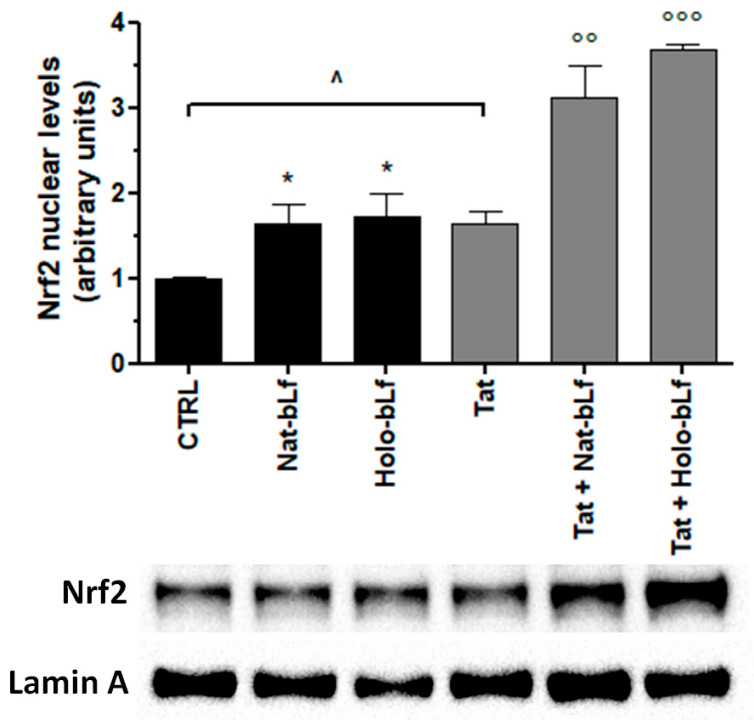
Analysis of Nrf2 nuclear translocation. Western blot and densitometric analysis of Nrf2 protein levels in nuclear extracts of U373 and U373-Tat cells after 4 h of treatment with 100 μg/mL of Nat-bLf or Holo-bLf. Data are calculated relative to the internal housekeeping gene (Lamin A) and are expressed as the means ± SEM. One-way ANOVA, followed by Tukey’s test, was used to determine significant differences. * *p* ≤ 0.05 vs. CTRL; °° *p* ≤ 0.01 and °°° *p* ≤ 0.001 vs. Tat; ^ *p* ≤ 0.05 between Tat and CTRL.

**Figure 3 ijms-24-07947-f003:**
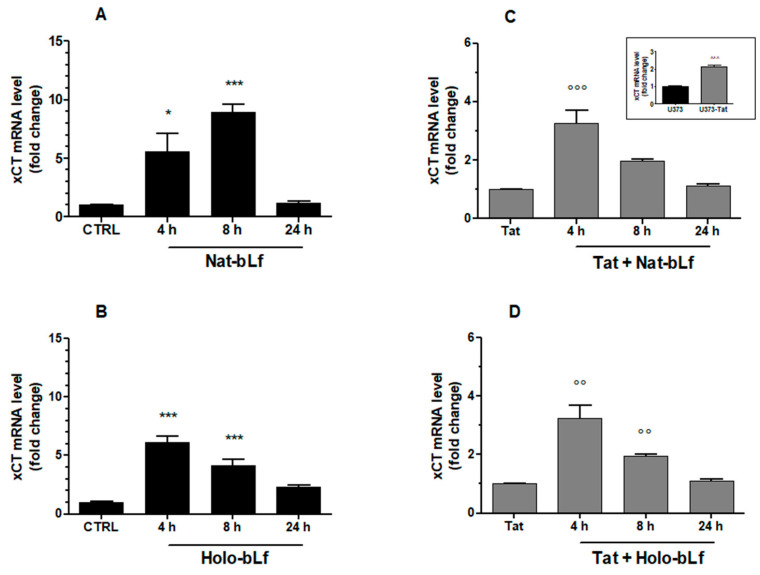
Analysis of System X_c_^−^ transcript levels. Real-Time qPCR analysis of System X_c_^−^ mRNA in U373 (**A**,**B**) and U373-Tat (**C**,**D**) cells after 4, 8, and 24 h of treatment with 100 μg/mL of Nat-bLf or Holo-bLf. Data are calculated relative to the internal housekeeping gene (GAPDH) and are expressed as the means ± SEM. One-way ANOVA, followed by Tukey’s test, was used to determine significant differences. * *p* ≤ 0.05 and *** *p* ≤ 0.001 vs. CTRL; °° *p* ≤ 0.01 and °°° *p* ≤ 0.001 vs. Tat; ^^^ *p* ≤ 0.001 between U373-Tat and U373 (inset).

**Figure 4 ijms-24-07947-f004:**
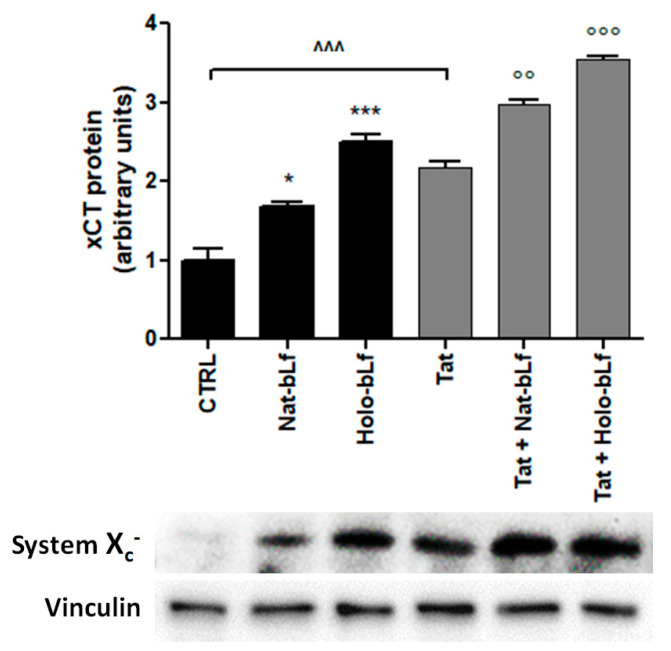
Western blot and densitometric analysis of System X_c_^−^ in U373 and U373-Tat cells treated with 100 μg/mL of Nat-bLf or Holo-bLf after 24 h. Data are calculated relative to the internal housekeeping gene (Vinculin) and are expressed as the means ± SEM. One-way ANOVA, followed by Tukey’s test, was used to determine significant differences. * *p* ≤ 0.05 and *** *p* ≤ 0.001 vs. CTRL; °° *p* ≤ 0.01 and °°° *p* ≤ 0.001 vs. Tat; ^^^ *p* ≤ 0.001 between Tat and CTRL.

**Figure 5 ijms-24-07947-f005:**
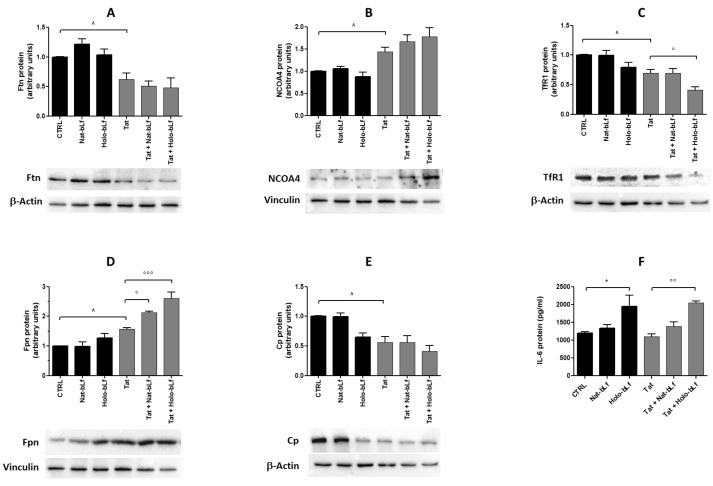
Western blot and densitometric analysis of Ftn (**A**), NCOA4 (**B**), TfR1 (**C**), Fpn (**D**), Cp (**E**), and ELISA quantification of IL-6 (**F**) in U373 and U373-Tat cells treated with 100 μg/mL of Nat-bLf or Holo-bLf for 48 h. Data are calculated relative to the internal housekeeping gene (β-Actin or Vinculin) and are expressed as the means ± SEM. One-way ANOVA, followed by Tukey’s test, was used to determine significant differences. * *p* ≤ 0.05 vs. CTRL; ° *p* ≤ 0.05, °° *p* ≤ 0.01 and °°° *p* ≤ 0.001 vs. Tat; ^ *p* ≤ 0.05 between Tat and CTRL.

**Figure 6 ijms-24-07947-f006:**
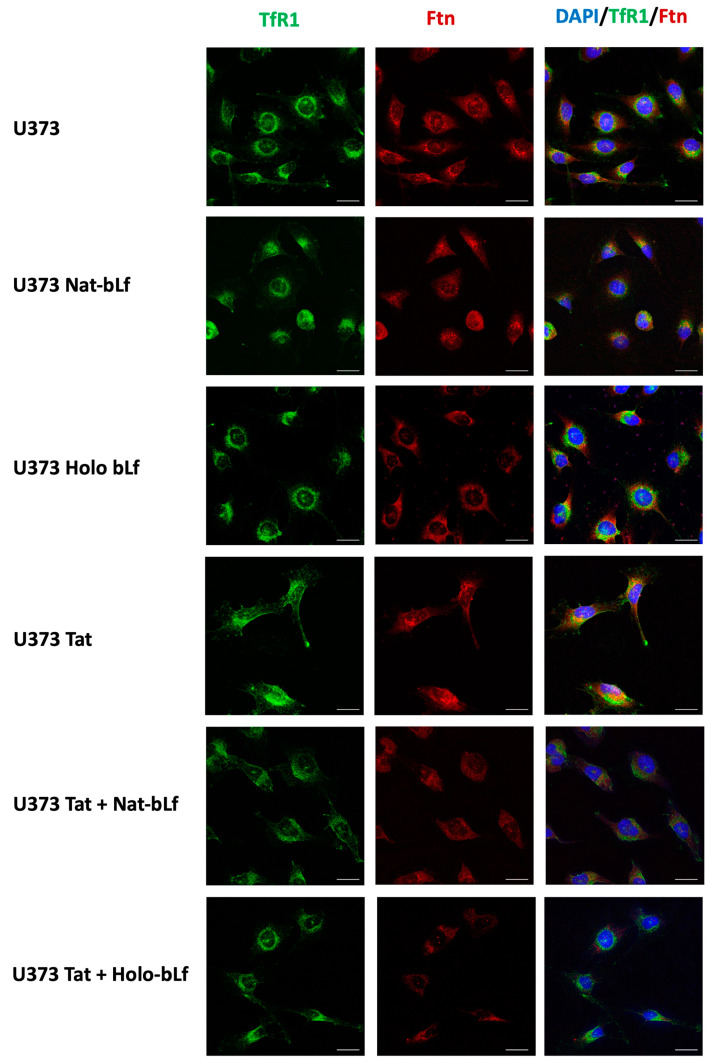
Immunofluorescence and confocal analysis for TfR1 (green) and Ftn (red) subcellular localization in U373 and U373-Tat cells treated with 100 μg/mL of Nat-bLf or Holo-bLf for 48 h. DAPI was used to stain nuclei (blue). Scale bar, 10 μm.

**Figure 7 ijms-24-07947-f007:**
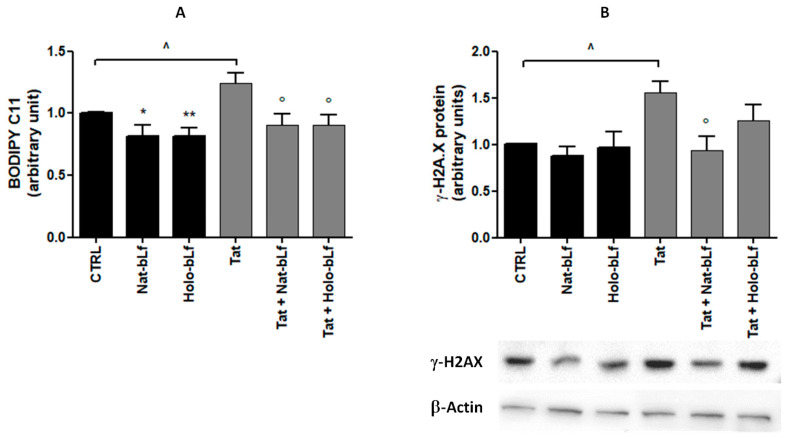
Analysis of lipid peroxidation by BODIPY assay (**A**) and Western blotting of γ-H2AX (**B**) in U373 and U373-Tat cells treated with 100 μg/mL of Nat- or Holo-bLf for 24 h. Data are calculated relative to the internal housekeeping gene (β-Actin) and are expressed as the means ± SEM (b). One-way ANOVA, followed by Tukey’s test, was used to determine significant differences. * *p* ≤ 0.05 and ** *p* ≤ 0.01 vs. CTRL; ° *p* ≤ 0.05 vs. Tat; ^ *p* ≤ 0.05 between Tat and CTRL.

**Figure 8 ijms-24-07947-f008:**
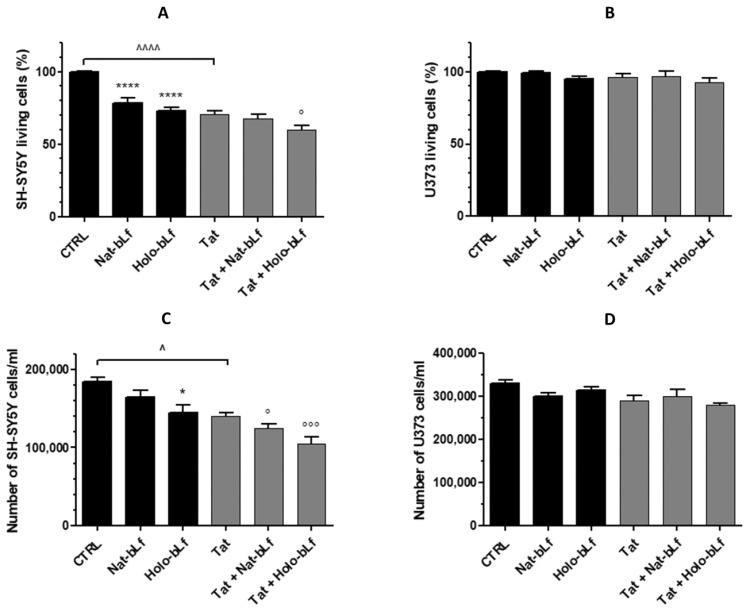
Cell viability measurement by MTT assay (**A**,**B**) and trypan blue exclusion assay (**C**,**D**) on co-cultures of SH-SY5Y and U373 or U373-Tat cells untreated or treated with 100 μg/mL Nat- or Holo-bLf for 24 h. The histograms in (**A**,**B**) show the percentage of living cells, and the rate of reduction was calculated by setting the control (CTRL) equal to 100%. One-way ANOVA, followed by Tukey’s test, was used to determine significant differences. * *p* ≤ 0.05 and **** *p* ≤ 0.0001 vs. CTRL; ° *p* ≤ 0.05 and °°° *p* ≤ 0.001 vs. Tat; ^ *p* ≤ 0.05 and ^^^^ *p* ≤ 0.0001 between Tat and CTRL.

**Figure 9 ijms-24-07947-f009:**
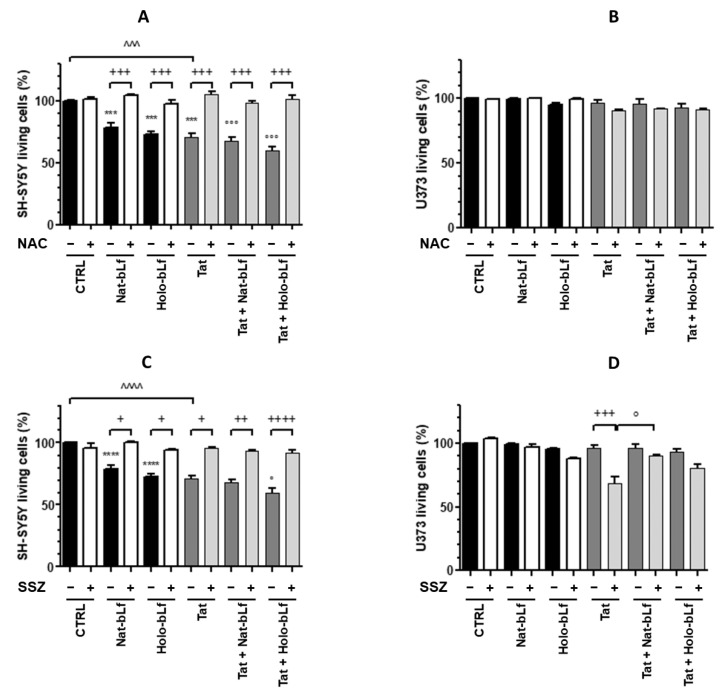
Cell viability measurement by MTT assay on co-cultures of SH-SY5Y and U373 or U373-Tat cells, pre-treated with 2 mM N-acetyl cysteine (NAC) (**A**,**B**) or 300 μM sulfasalazine (**C**,**D**), untreated or treated with 100 μg/mL Nat- or Holo-bLf for 24 h. The histograms show the percentage of living cells, and the rate of reduction was calculated by setting the control (CTRL) equal to 100%. One-way ANOVA, followed by Tukey’s test, was used to determine significant differences. *** *p* ≤ 0.001 and **** *p* ≤ 0.0001 vs. CTRL; ° *p* ≤ 0.05 and °°° *p* ≤ 0.001 vs. Tat; ^^^ *p* ≤ 0.001 and ^^^^ *p* ≤ 0.0001 between Tat and CTRL; + *p* ≤ 0.05, ++ *p* ≤ 0.01, +++ *p* ≤ 0.001 and ++++ *p* ≤ 0.0001 between treatment with SSZ or NAC and without SSZ or NAC.

**Figure 10 ijms-24-07947-f010:**
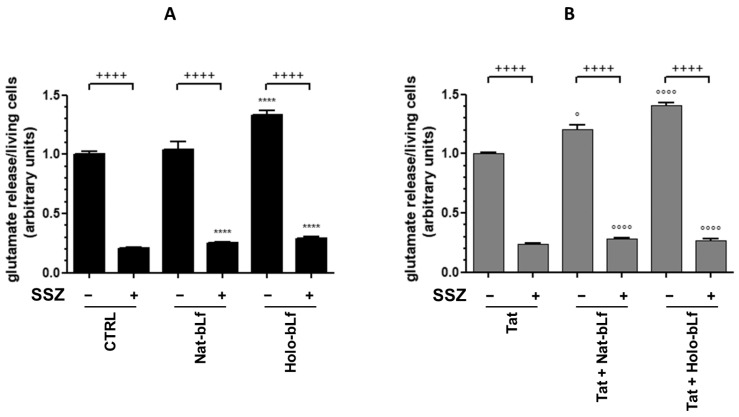
Measurement of glutamate release in co-culture supernatants of SH-SY5Y with U373 (**A**) or U373-Tat (**B**) cells untreated or treated with 300 μM sulfasalazine and 100 μg/mL Nat- or Holo-bLf for 24 h. One-way ANOVA, followed by Tukey’s test, was used to determine significant differences. **** *p* ≤ 0.0001 vs. CTRL; ° *p* ≤ 0.05 and °°°° *p* ≤ 0.0001 vs. Tat; ++++ *p* ≤ 0.0001 between treatment with or without SSZ.

**Figure 11 ijms-24-07947-f011:**
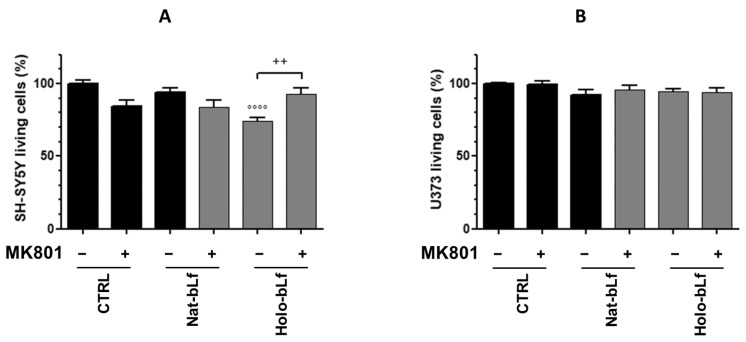
Cell viability measurement by MTT assay on SH-SY5Y cells (**A**) and U373-Tat cells (**B**) in co-cultures, untreated or treated with 10 μM MK801 and 100 μg/mL Nat- or Holo-bLf for 24 h. The histograms show the percentage of living cells, and the rate of reduction was calculated by setting the control (CTRL) equal to 100%. One-way ANOVA, followed by Tukey’s test, was used to determine significant differences. °°°° *p* ≤ 0.0001 vs. CTRL; ++ *p* ≤ 0.01 between treatment with or without MK801.

**Figure 12 ijms-24-07947-f012:**
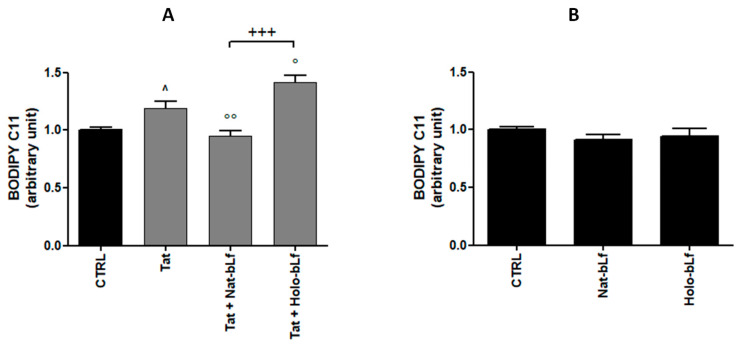
Analysis of lipid peroxidation by BODIPY assay in co-cultures of SH-SY5Y with U373-Tat cells (**A**) or in SH-SY5Y monocultures (**B**) untreated or treated with 100 μg/mL Nat- or Holo-bLf for 24 h. One-way ANOVA, followed by Tukey’s test, was used to determine significant differences. ° *p* ≤ 0.05 and °° *p* ≤ 0.01 vs. Tat; ^ *p* ≤ 0.05 between Tat and CTRL; +++ *p* ≤ 0.001 between treatment with Nat- and Holo-bLf.

## Data Availability

The data presented in this study are available in the current article.

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
