# Peer review of "Iron Saturation Drives Lactoferrin Effects on Oxidative Stress and Neurotoxicity Induced by HIV-1 Tat"

_ijms, 2023, doi:10.3390/ijms24097947_

Round 1

Reviewer 1 Report

Summary:

The authors examine the effectiveness of bovine lactoferrin (bLf) in mitigating oxidative stress in astrocytes expressing the HIV-1 protein Tat. Results show that both native (Nat-bLf) and iron-saturated (Holo-bLf) forms of bLf increase antioxidant response and reduce ROS-mediated damage in astrocytes by up-regulating System Xc- and ferroportin through the Nrf2 pathway. Both forms of bLf also restore the physiological internalization of Transferrin Receptor 1, which is important for iron uptake. However, Holo-bLf exacerbates Tat-induced excitotoxicity on neurons through System-Xc- up-regulation, emphasizing the significance of iron in bLf's biological functions. Although certain parts of this study could use improvement, the general concept behind it is quite sound.

Major:

1.     Line 159: Can an explanation be provided for why the peak was observed at 4-8 hours?

2.     Figure 6 (confocal) shows that "U373 Tat + Nat-bLf" has a lower TfR1 level than "U373 Tat + Holo-bLf," whereas in Figure 5C (western blot), "U373 Tat + Nat-bLf" has a higher TfR1 level than "U373 Tat + Holo-bLf." Is there a possible explanation for this difference?

3.     Further discussions on the comparison of Holo-bLf and Nat-bLf, particularly regarding their differing performances, such as shown in Figure 5C, would be helpful.

4.     The immunofluorescence and confocal analysis in Figure 6 appear dim and blurry. Please adjust the images to improve their brightness and clarity.

5.     Please provide more details on the bodipy assay in the Materials and Methods section.

Minor:

1.     Could the authors confirm if all experiments were repeated independently three times? If so, please indicate in the figure legends or Materials and Methods section.

2.     Please use consistent terminology, such as "sub-cellular" (line 130) vs. "subcellular" (line 225).

3.     It is recommended to use italics for the p value.

The language of this paper is clear, concise, and straightforward.

Author Response

We thank the Reviewer for their helpful comments. The paper has been amended accordingly and a point-to-point reply follows.

Major:

  1. Line 159: Can an explanation be provided for why the peak was observed at 4-8 hours?

The peak observed at 4-8 h for system Xc mRNA expression is fully compatible with the activation of Nrf2 which was detected at 4 hours, considering that translocation of the transcription factor and mRNA expression occur sequentially. For this reason, system Xc mRNA expression was not analyzed at shorter times, and system Xc protein was measured at 24 hours.

  1. Figure 6 (confocal) shows that "U373 Tat + Nat-bLf" has a lower TfR1 level than "U373 Tat + Holo-bLf," whereas in Figure 5C (western blot), "U373 Tat + Nat-bLf" has a higher TfR1 level than "U373 Tat + Holo-bLf." Is there a possible explanation for this difference?

As pointed out by both this Reviewer and by Reviewer 3, the quality of Figure 6 was not adequate. A more thorough review of the IF data showed more similar signals in Nat- or Holo-bLf-treated U373 Tat cells. The figure has been therefore improved. The apparent discrepancy between microscopy and western blot results can also be due, at least in part, to the fact that signals are more diffuse in “U373 Tat + Nat-bLf” TfR1 than in “U373 Tat + Holo-bLf”, which makes it difficult to quantify relative intensities by visual inspection.

  1. Further discussions on the comparison of Holo-bLf and Nat-bLf, particularly regarding their differing performances, such as shown in Figure 5C, would be helpful.

Along with the already present comparison between Nat- and Holo-bLf performances towards anti-oxidant response, inflammation, DNA damage and neurotoxicity, a brief note about the Holo-bLf effect on TfR1 expression has been added (lines 380-383).

  1. The immunofluorescence and confocal analysis in Figure 6 appear dim and blurry. Please adjust the images to improve their brightness and clarity.

As also explained in point 2 above, Figure 6 has been adjusted as requested.

  1. Please provide more details on the bodipy assay in the Materials and Methods section.

Details on the Bodipy assay are now reported in the Materials and Methods as section 4.11 (lines 580-591).

Minor:

  1. Could the authors confirm if all experiments were repeated independently three times? If so, please indicate in the figure legends or Materials and Methods section.

The number of independent repetitions is now reported in the Materials and Methods section (lines 493-494).

  1. Please use consistent terminology, such as "sub-cellular" (line 130) vs. "subcellular" (line 225).

Done.

  1. It is recommended to use italics for the p value.

Done.

Reviewer 2 Report

In the submitted work, the Authors present several pieces of intriguing data pertaining to a deleterious neuropathological exacerbation in HIV infection.   Reviewer was likewise interested to see described the role that Holo-bLF 30 plays in modulating System-Xc-, particularly given the pathognomonic influence glutamate exerts.  Given the subject matter, Reviewer was mildly surprised to find no mention that lactoferrin may offer therapeutic benefit, specifically in preventing the onset of amyloidosis (CITE -- https://pubmed.ncbi.nlm.nih.gov/18717818/)

It appears that a key feature of the submitted work discusses the importance of inflammation in the onset and progression of phenotype.  Yet, it also seems the Authors only lightly touch on the specifics by articulating very few relevant markers, e.g. IL6.  Given the critical role such triggers play here, would not a somewhat more broad interrogation of such markers, than those mentioned, prove enlightening?  

On another topic, while Reviewer agrees that bLf regulatory oversight may prove beneficial -- the Authors dramatically overreach by appearing to lump "all" nutraceuticals together via their paraphrased suggestion that [nutraceuticals be tightly regulated].  As a minor correction, in line 147 "about" is not an appropriate scientific term and should be substituted accordingly.  

Paper could benefit from a 'once over' grammatical review.  

Author Response

We thank the Reviewer for their helpful comments. The paper has been amended accordingly and a point-to-point reply follows.

In the submitted work, the Authors present several pieces of intriguing data pertaining to a deleterious neuropathological exacerbation in HIV infection.  Reviewer was likewise interested to see described the role that Holo-bLF 30 plays in modulating System-Xc-, particularly given the pathognomonic influence glutamate exerts.  Given the subject matter, Reviewer was mildly surprised to find no mention that lactoferrin may offer therapeutic benefit, specifically in preventing the onset of amyloidosis (CITE -- https://pubmed.ncbi.nlm.nih.gov/18717818/)

We included this observation in the Discussion section (lines 433-436; lines 449-450) and the suggested citation has been added as ref. 56.

It appears that a key feature of the submitted work discusses the importance of inflammation in the onset and progression of phenotype.  Yet, it also seems the Authors only lightly touch on the specifics by articulating very few relevant markers, e.g. IL6.  Given the critical role such triggers play here, would not a somewhat more broad interrogation of such markers, than those mentioned, prove enlightening?  

We agree with the Referee that inflammation represents a hallmark of HAND, playing a pivotal role in the onset and development of this pathology. However, different cell types contribute to inflammation in HAND, including infiltrated monocytes/macrophages and microglia. The present study is focused on the role of the two main cytokines involved in dysregulation of iron metabolism, namely IL-6 and IL-1b, in exacerbating Tat-induced neurotoxicity. For this reason, we believe that a deeper investigation on other inflammatory markers would be beneficial in the framework of future studies preferentially involving animal models.     

On another topic, while Reviewer agrees that bLf regulatory oversight may prove beneficial -- the Authors dramatically overreach by appearing to lump "all" nutraceuticals together via their paraphrased suggestion that [nutraceuticals be tightly regulated].  As a minor correction, in line 147 "about" is not an appropriate scientific term and should be substituted accordingly.  

Corrections have been carried out according to the Reviewer’s suggestions (line 147; lines 437-439).

Paper could benefit from a 'once over' grammatical review. 

Done.

Reviewer 3 Report

Authors identified an interesting finding that he lactoferrin can exacerbate the HIV protein Tat-induced toxicity in human glial cell line. Here are my concerns:

Major: 

1, All the experiments in this study used the U373 cell line. Cell lines can behave very differently compared with normal cells. At least the most important phenotypic experiments should be performed using isolated glial cells from mouse or rat.

2, The original uncropped Western Blot bands should be provided.

Minor:

1, The TfR1 and Cp bands in Figure 5C nd 5E should be replaced with better bands without exaggerated cropping.

2, The IF intensity of the different panels in Figure 6 should be adjusted to the similar level.

Minor language improvement will be needed.

Author Response

We thank the Reviewer for their helpful comments. The paper has been amended accordingly and a point-to-point reply follows.

Major: 

1, All the experiments in this study used the U373 cell line. Cell lines can behave very differently compared with normal cells. At least the most important phenotypic experiments should be performed using isolated glial cells from mouse or rat.

This observation is certainly pertinent. However, the sole use of normal mouse/rat glial cells, as suggested by the referee, would not solve the problem, as the most relevant experiments in our manuscript make use of cocultures of astrocytes/neurons. Cocultures of primary cells is a challenging task as normal neurons and astrocytes would have to be obtained from neural progenitors or induced pluripotent stem cells (iPSC). We feel that in this paper we provide relevant and sufficient evidence on cellular mechanisms at a molecular level, although our study was performed on cancer cell line models. In our opinion, this represents a very important point and confers a physiological relevance to our study.

We certainly agree with the referee that investigation on “normal” cells can be a very important topic for our future research and we are currently putting our efforts in this field. Apart from considerations above, at this stage executions times would be not compatible with the editor's request to revise the paper within 10 days.

2, The original uncropped Western Blot bands should be provided.

Following IJMS policy, the original uncropped Western blots had been already sent to the Editor for preliminary check and approval.

Minor:

1, The TfR1 and Cp bands in Figure 5C and 5E should be replaced with better bands without exaggerated cropping.

Cropping of bands in Figure 5C and 5E has been improved.

2, The IF intensity of the different panels in Figure 6 should be adjusted to the similar level.

Figure 6 has been adjusted (see also Reviewer 1).

Round 2

Reviewer 1 Report

Please make sure to check for any formatting errors on line 453 before publishing. Thanks.

N/A

Author Response

Thank the Reviewer for the comment, however, the formatting on line 453 is fine to us.

Reviewer 3 Report

Authors handled all my concerns well. I don't have further questions.

Author Response

Thank the Reviewer for the comment